# Fly-Ash-Based Geopolymers Reinforced by Melamine Fibers

**DOI:** 10.3390/ma14020400

**Published:** 2021-01-15

**Authors:** Barbara Kozub, Patrycja Bazan, Dariusz Mierzwiński, Kinga Korniejenko

**Affiliations:** Institute of Materials Engineering, Faculty of Material Engineering and Physics, Cracow University of Technology, Jana Pawła II 37, 31-864 Cracow, Poland; barbara.kozub@pk.edu.pl (B.K.); patrycja.bazan@pk.edu.pl (P.B.); dariusz.mierzwinski@pk.edu.pl (D.M.)

**Keywords:** geopolymer composite, fly ash, melamine, compressive strength, flexural strength, thermal radiation changes

## Abstract

This paper presents the results of research on geopolymer composites based on fly ash with the addition of melamine fibers in amounts of 0.5%, 1% and 2% by weight and, for comparison, without the addition of fibers. The melamine fibers used in the tests retain their melamine resin properties by 100% and are characterized by excellent acoustic and thermal insulation as well as excellent filtration. In addition, these fibers are nonflammable, resistant to chemicals, resistant to UV radiation, characterized by high temperature resistance and, most importantly, do not show thermal-related shrinking, melting and dripping. This paper presents the results of density measurements, compressive and flexural strength as well as the results of the measurement of thermal radiation changes in samples subjected to a temperature of 600 °C. The results indicate that melamine fibers can be used as geopolymer reinforcement. The best result was achieved for 0.5% by weight amount of reinforcement, approximately 53 MPa, compared to 41 MPa for a pure matrix. In the case of flexural strength, the best results were obtained for the samples made of unreinforced geopolymer and samples with the addition of 0.5% by weight of melamine fibers, which were characterized by bending strength values above 9 MPa, amounting to 10.7 MPa and 9.3 MPa, respectively. The thermal radiation measurements and fire-jet test did not confirm the increasing thermal and fire resistance of the composites reinforced by melamine fiber.

## 1. Introduction

Almost three decades ago, the French scientist Joseph Davidovits introduced the term “geopolymers” for aluminosilicate polymers formed in an alkaline environment [1,2,3]. Most geopolymer synthesis methods come down to one process, in which comminuted dried pozzolanic material (such as metakaolin [4,5,6], fly ash [7,8,9,10] or blast furnace slag [11,12]) is mixed with an aqueous solution of a suitable silicate (e.g., sodium or potassium silicate) with the addition of a strong base, usually concentrated sodium or potassium hydroxide. The resulting paste behaves similarly to the cement paste. It solidifies to a hard mass within a few hours; however, it is made without the use of Portland cement [13,14,15].

Geopolymers, due to their unique properties, have aroused the interests of not only scientific communities but also entrepreneurs interested in new competitive technologies over the past several years. Geopolymer-based products are increasingly beginning to appear on the market. In recent years, geopolymers have been considered some of the most interesting and promising building materials [16,17]. There are many types of geopolymers, and they have interesting applications. One of the most promising ways geopolymers are implemented is in fireproof and fire-resistant products such as shields able to withstand very high temperatures [18,19,20]. Numerous studies in the literature mainly present geopolymers as alternatives to cement for structural applications. Recently, several publications have appeared indicating their potential as thermal barriers [16,17]. These studies indicate that geopolymers can achieve high thermal stability and fire resistance. Moreover, the production technology itself can be classified as economical and safe, both for people and the environment. This presents a significant advantage over conventional thermal insulation materials, the properties of which, at a temperature of around 80 °C, are significantly deteriorated; what is worse, they often emit dangerous emissions and toxic gases, posing a threat to human health and life [19,20,21].

A limitation to the wide application of geopolymer materials is their relatively low brittle behavior (crack resistance) [22,23]. Currently, one of the most important research areas concerns the improvement of these mechanical properties [24,25]. A well-known solution is the production of fiber-reinforced composite materials. Therefore, numerous studies have recently been conducted on the reinforcement possibilities of geopolymer materials with various types of fibers [20,26]. These studies mainly focus on the use of short fibers for the reinforcement of composites with a geopolymer matrix.

Polymer fibers are the most frequently used group of fibers as reinforcing additives [25]. The main objective of the addition of these types of fibers is improving mechanical properties, in particular bending strength [24,27], and reducing the propagation of microcracks in materials [23,28]. Other benefits that can be achieved as a result of introducing chemical fibers into geopolymers, depending on their type, may include an increase in fire resistance [24,27] or a decrease in the thermal conductivity coefficient. The main advantages of chemical, polymer fibers are their higher strength properties and repeatability compared to natural fibers [25,29]. Additional features desirable for particular applications may include decreasing the weight of geopolymer composites with the addition of polymer fibers [24,25].

The background for this work was literature research showing the novelty aspects of the planned research. The available literature does not present research using as many possible fillers as reinforcement for geopolymer matrices, such as melamine fibers [25,29]. Research using melamine has been conducted with organic polymers. The results suggest that this kind of reinforcement could provide the composites with better thermal and fire resistance than using other polymer fibers, including glass fibers [30,31,32].

The present work attempts to design and test a new composite with better mechanical properties and increased thermal resistance compared to pure matrices [33,34]. This type of composite could be used as thermal insulation in civil engineering structures. This paper presents the results of research on geopolymer composites with a matrix based on fly ash and with the addition of melamine fibers in the amount of 0.5%, 1% and 2% by weight. The effect of melamine addition on selected properties of fly ash matrix geopolymers is then examined.

## 2. Materials and Methods

### 2.1. Materials and Preparation of Samples

The geopolymer matrix was made of F class fly ash from the Skawina Combined Heat and Power Plant (Skawina, Poland) and fine-grained, saturated-surface, dry construction sand (the surfaces of the sand particles are “dry” but the voids between the particles are saturated with water and there is no surface absorption) in the ratio 1:1. The type of fly ash used in this research consists mainly of silica and aluminum oxide and contains less than 4% calcium oxide. The exact percentage of the individual phases in the fly ash used can be found in Table 1 [29]. This type of fly ash is characterized by specific physical and chemical properties to support the process of geopolymerization [35,36].

Melamine fibers (smartMELAMINE^®®^, the smart polymer GmbH, Rudolstadt, Germany) were used as reinforcement for the geopolymer composites. Melamine was added in an amount of 0.5% (sample marked as 0.5%MF), 1% (1%MF) or 2% (2%MF) by weight of dry components (Table 2). A reference sample based on the matrix material without any additives (samples marked as 0%MF) was made for comparison.

In this study, a nonwoven fabric (consisting of very fine melamine fibers), with a texture resembling wadding was used, cut into pieces about 5 mm long (Figure 1). The melamine fibers used in the tests retain their melamine resin properties and are characterized by excellent acoustic and thermal insulation as well as excellent filtration. In addition, these fibers are nonflammable, resistant to chemicals, resistant to UV (ultraviolet) radiation, characterized by high temperature resistance and, most importantly, they do not show thermal-related shrinking, melting and dripping [37].

As an alkaline activator, a 10-molar (10 M) sodium hydroxide solution and a sodium water glass R-145 (with a molar module of 2.5 and a density of about 1.45 g/cm^3^) combined in the ratio of 1:2 were used. The alkaline solution was prepared by pouring an aqueous solution of sodium silicate into the flakes of the technical sodium hydroxide dissolved in water. The solution was then mixed thoroughly and allowed to equilibrate until a constant concentration and ambient temperature were reached, which took nearly 2 h. The W/C ratio was selected according to a previous experiment [29]. To prepare geopolymer masses, the fly ash, construction sand, melamine fibers and alkaline solution were mixed to achieve a homogeneous paste for about 20 min in a low-speed mixing machine. The solid ingredients were added to fire and followed by liquid. The melamine fibers were dispersed randomly in the geopolymer matrix, and the homogeneous dispersion of the fibers was assessed visually.

In the next step, the prepared masses were poured into plastic molds, which were subjected to vibration on a vibrating table in order to remove air bubbles. The tightly closed molds were then heated at a temperature of 75 °C in a laboratory dryer for 24 h. After 28 days, the samples were unmolded, tested and stored in laboratory conditions (temperature ca. 20 °C, relative humidity ca. 50%).

### 2.2. Methodology

#### 2.2.1. Density

Before carrying out the strength tests, the density of the samples was determined using the geometric method. The density for each of the analyzed geopolymer compositions was determined as the average of the measurements for four samples. The dimensions of the samples were measured with a laboratory caliper with a measuring accuracy of 0.01 mm, and the weight of the samples was determined using the RADWAG PS 200/2000.R2 laboratory precise analytical balance (maximum load: 200/2000 g; reading accuracy: 0.001/0.01 g). Because the composites were not characterized by significant porosity (solid material without voids), the calculations were carried out for solid, nonporous materials.

#### 2.2.2. Strength Tests

Due to the lack of separate standards for geopolymer materials, the compressive strength tests were carried out in accordance with the procedure described in the standard for concrete, EN 12390-3 (“Testing hardened concrete. Compressive strength of test specimens”). The tests were carried out on the Matest 3000 kN (Matest, Treviolo, Italy) universal strength testing machine with a speed of 0.05 MPa/s. For each analyzed chemical composition of geopolymer composites, four cubic samples with dimensions (approximately) 50 mm × 50 mm × 50 mm were made and tested for each composition. The dimensions of the samples comply with the EN 12390-3 standard.

Flexural strength tests were also performed. As in the case of the compressive strength tests, due to the lack of standards for geopolymers, the standard for concrete EN 12390-5 (“Testing hardened concrete. Flexural strength of test specimens”) was used. The tests were also carried out on the Matest 3000 kN universal testing machine with a speed of 0.05 MPa/s. For each analyzed geopolymer composites, four prismatic samples with dimensions (approximately) of 50 mm × 50 mm × 200 mm were made and tested for each composition, with the distance between the support points equal to 150 mm. The dimensions of the samples comply with the EN 12390-5 standard.

#### 2.2.3. Thermal Radiation

Four plates were subjected to thermal radiation measurement. Each sample, in the form of a plate with dimensions (approximately) 50 mm × 100 mm × 150 mm, was placed in a silicate chamber (electric) furnace in such a way as to constitute an insulating element, according to the scheme presented in Figure 2. The sealing element between the sample and the walls of the furnace was an element made of an insulating material able to withstand temperatures up to about 1500 °C.

The measurement of the change in thermal radiation was performed on the surface of the sample constituting the external part of the measuring system (Figure 3).

The measurement was performed with a FLIR thermal imaging camera with a field of view (FOV) ≥ 38°, thermal sensitivity < 70 mK, measured infrared wavelength range in the range of 7–14 µm and pixel size < 15 µm. The thermal camera was placed 1.5 m from the tested system. The measurement of thermal radiation was made pointwise from the center of the sample surface and was performed with a frequency of 60 s for a period of 1 h.

#### 2.2.4. Fire-Jet Test

Two plates were subjected to a heating test similar to the fire-jet test: 0%MF and 0.5%MF. The samples were chosen after the results of the thermal radiation investigation and the analysis of the results of the mechanical strengths, where the best results were achieved for the composite with the addition of 0.5% by weight of melamine fibers. The gas emitter was an oxygen–acetylene burner, which was set at a distance from the sample in such a way that the temperature on the surface did not exceed 1500 °C (Figure 4).

The test lasted until cracks were observed on the side of the plate, which was not exposed to the exposure of combustion gases or “open fire” (Figure 5).

The temperature measurement on the side opposite to the heat source was made with a thermal imaging camera every 5 min. The measurement was performed with a FLIR thermal imaging camera with a field of view (FOV) ≥ 38°, thermal sensitivity < 70 mK, measured infrared wavelength range in the range of 7–14 µm and pixel size < 15 µm. The use of this method guaranteed the continuity of the material surface and did not generate additional stresses.

## 3. Results and Discussion

### 3.1. Density

The results of the compressive and flexural strength tests are shown in Figure 6.

Before carrying out the strength tests, in order to determine the density of the tested geopolymers, the samples intended for the compressive strength tests were measured and weighed. The obtained results of the geopolymer composites’ density are shown in Figure 6. The density value for the material without reinforcement was 1.81 g/cm^3^. In the case of the geopolymer composite with the addition of 0.5% by weight of melamine fibers, the density value was almost on the same level. The addition of the melamine reinforcement in the amount of 1% and 2% by weight allowed for a slight decrease in the density of composites in comparison to the reference sample, which were, respectively, 1.75 g/cm^3^ and 1.68 g/cm^3^. However, the decrease in these values is directly related to the lower density of the added melamine fibers equal to about 1.6 g/cm^3^. The achieved results are in accordance with other research provided on geopolymer matrices reinforced by polymer fibers given in the literature [25,29]. The density of geopolymers is usually between 1.5 g/cm^3^ and 2.0 g/cm^3^ and is strictly related to the used raw materials such as fly ash. The addition of polymer fibers decreases the density of the composites because of their low weight.

### 3.2. Strength Tests

The results of the compressive and flexural strength tests for all investigated materials are shown in Figure 7. The results of both unreinforced and fiber-reinforced specimens are presented.

The compressive strength for the samples made of the reference material (with pure geopolymer matrix) reached a value of about 41 MPa. Better results were obtained for geopolymer composites with the addition of 0.5% and 1% by weight of melamine fibers, for which the compressive strength was approximately 53 MPa and 45 MPa, respectively (Figure 7a). In the case of the composite with the addition of 2% by weight of melamine, the determined compressive strength dropped significantly to a value of about 25 MPa, which is equal to about 60% of the compressive strength of the sample with the pure geopolymer matrix. It was most likely connected to high fiber content because by increasing their content in the composite, the workability of the geopolymer mass decreased significantly. In the case of the sample with the addition of 2% by weight of melamine reinforcement, the workability of the mass was the worst, and the obtained samples did not maintain the correct dimensions; the mass did not spread evenly, and the distribution of the fibers was most likely not homogeneous. The workability of the mass could also be influenced by the fact that the melamine fibers were characterized by high absorption of the activating solution. The highest compressive strength was obtained for the geopolymer composite with the addition of 0.5% by weight of melamine fibers, and its value was about 24% higher than the compressive strength value for samples made of a pure geopolymer matrix. This result seems to be promising for future applications and is coherent with the results of other research [25,26,38,39,40,41,42].

During the analysis of the results obtained from the bending tests, it can be observed that the higher the melamine fiber content in the composite, the lower the obtained bending strength values. As a result, only samples made of the unreinforced geopolymer and samples with the addition of 0.5% by weight of melamine fibers were characterized by bending strength values above 9 MPa, amounting to 10.7 MPa and 9.3 MPa, respectively (Figure 7b). The results show the nontypical behavior of the geopolymer composite reinforced by polymer fibers. The flexural strength usually increased [25], but an exception has been confirmed by other research teams. This behavior could be explained by a material imperfection that caused decreasing flexural strength or an improper interfacial mechanism between the fibers and the matrix [43,44,45].

### 3.3. Thermal Radiation

The graph in Figure 8 shows the temperature change of a geopolymer plate with different melamine contents measured on the outer surface of the plate as an insulating element (thermal barrier).

The green and blue curves corresponding to the composite containing 0.5% and 1% by weight of melamine in relation to the total weight of the geopolymer matrix do not show significant changes in the emission of thermal radiation with increasing temperature in the furnace. Compared to the geopolymer matrix without melamine addition, both curves are characterized by a lower temperature on average by about 4 °C.

When comparing the pure geopolymer matrix (red curve) and the geopolymer matrix containing 2% by weight of melamine (purple curve), a lower temperature can be observed on the surface of the sample containing melamine in the initial heating stage. This situation changes after approximately 25 min of temperature rise in the furnace. After this time, the temperature measured on the outer surface of the plate shows a higher temperature by more than 4 °C compared to the pure geopolymer matrix.

A comparison of the temperature difference between the pure geopolymer matrix and the matrix with variable melamine content is shown in Figure 9.

Figure 10 shows exemplary photos of the sample surface before and after exposure to a temperature of 600 °C. Discoloration and numerous surface cracks could be observed on the surface of the samples.

### 3.4. Fire-Jet Test

Figure 11 presents the temperature changes measured on the outer surface of the plate (opposite to open fire) for geopolymer plates with different melamine contents.

Analyzing the obtained curves shown in Figure 11, a higher temperature on the tested surface of the geopolymer plate with the addition of melamine fibers can be observed, which may be the result of a marked degradation of the surface after exposure to open fire. The visual observations after the test confirm that the sample with melamine was clearly degraded. It is clearly visible also as a temperature difference between the sample with melamine and the reference sample (with pure geopolymer matrix).

## 4. Discussion

This paper presents geopolymer composites reinforced with melamine fibers. The density measurement results are as expected showing that, with the addition of the melamine reinforcement, there is a slight decrease in the density of the composites. This is in line with other research conducted on polymeric fibers that have a density lower than the geopolymer matrix [25,29].

The compressive strength test results indicate that melamine fibers can be used as geopolymer reinforcement. The best result was achieved for a 0.5% by weight amount of reinforcement, approximately 53 MPa, compared to 41 MPa for a pure matrix. The geopolymer composites with the addition of 1% and 2% by weight of melamine fibers displayed compressive strength values of approximately 45 MPa and 25 MPa, respectively. The obtained results are in line with those of other research teams. In the literature, the reported compressive strength values for geopolymers based on fly ash are usually between 20 and 80 MPa [25]. The obtained values are strongly dependent on fly ash quality [35,36]. The decrease in the compressive strength for the composite with 2% of fibers was caused by the worse workability of the geopolymer paste. This made the mixing process less efficient, which resulted in the formation of agglomerates of fibers in the material. Investigations conducted with the addition of other fibers confirmed that above a certain amount of the fiber, the mechanical properties of geopolymer composites decreases [25,26,38]. The overall tendency of increasing the compressive strength is not obvious, and much research shows that the compressive strength of geopolymer composites tends to decrease with an increasing amount of added polymer fibers [39,40,41,42].

However, the main reason for polymer fiber addition is not to increase the compressive strength but to improve the flexural strength and the behavior of the fracture tongues. It is expected that both of these properties will increase with the fiber amount [24,25]. The flexural strength provided in this research behaves quite differently. The best results were obtained for the samples made of an unreinforced geopolymer and samples with the addition of 0.5% by weight of melamine fibers, which were characterized by bending strength values above 9 MPa, amounting to 10.7 MPa and 9.3 MPa, respectively. The obtained results are highly comparable with the range of values for geopolymers provided by other research teams. In the literature, the flexural strength values for geopolymers based on fly ash are usually reported between 3 and 10 MPa [25]. The achieved results are strongly dependent on fly ash quality [35,36]. The overall tendency shows that flexural strength decreases with fiber addition. This behavior is not typical for polymeric fibers, and flexural strength typically increases by about 50% [25]. However, some studies demonstrate that it decreases or does not significantly change. Authors usually explain this by voids in the material, fiber agglomeration or the improper interfacial mechanism between the fibers and the matrix [43,44,45].

The thermal radiation measurements and fire-jet test do not confirm the increasing thermal and fire resistance of the composites reinforced by melamine fiber. The literature shows geopolymers with the addition of various fibers can be excellent materials for thermal- and fire-resistant applications [19,46]. The obtained results of the fire-jet test could be compared with the sandwich panels investigated by a team from the Czech Republic [18,47], where a geopolymer was applied as one of the layers. These composites resisted flames for more than 13 min [18,47]. However, the samples with melamine fibers degraded significantly faster than those without reinforcement during the fire test. This may be due to the lack of additional surface coverage and the action of melamine fibers, which have a much lower melting point than the matrix material (“hot spots”).

## 5. Conclusions

This paper presents geopolymer composites reinforced with melamine fibers. The results indicate that the inclusion of 0.5% by weight of melamine fibers significantly enhanced compressive strength and slightly decreased flexural strength in comparison to the unreinforced geopolymer based on fly ash, which means that melamine fibers can be successfully used to increase resistance to the axial compression of geopolymer composites. The obtained results seem to be promising for future applications, showing that it is possible to produce composites of reasonable properties from industrial wastes (fly ash) and melamine fibers. It should be noted that when melamine fibers are used as reinforcement for geopolymer composites, their amount should not exceed 0.5% by weight as exceeding this value may result in the deterioration of the workability of the geopolymer’s mass and a significant reduction in compressive strength and bending strength.

The obtained differences of the outer surface temperature of the tested plates at 4 °C in favor of composites with the addition of 0.5% and 1% of melamine fibers in relation to unreinforced geopolymer seem to be small, and further research should be carried out in order to achieve temperature stabilization (steady state) on the outer surface of the tested plate.

Practical applications require further testing to optimize the mechanical properties of the composites as well as investigating other properties such as water absorption and resistance in various environments.

## Figures and Tables

**Figure 1 materials-14-00400-f001:**
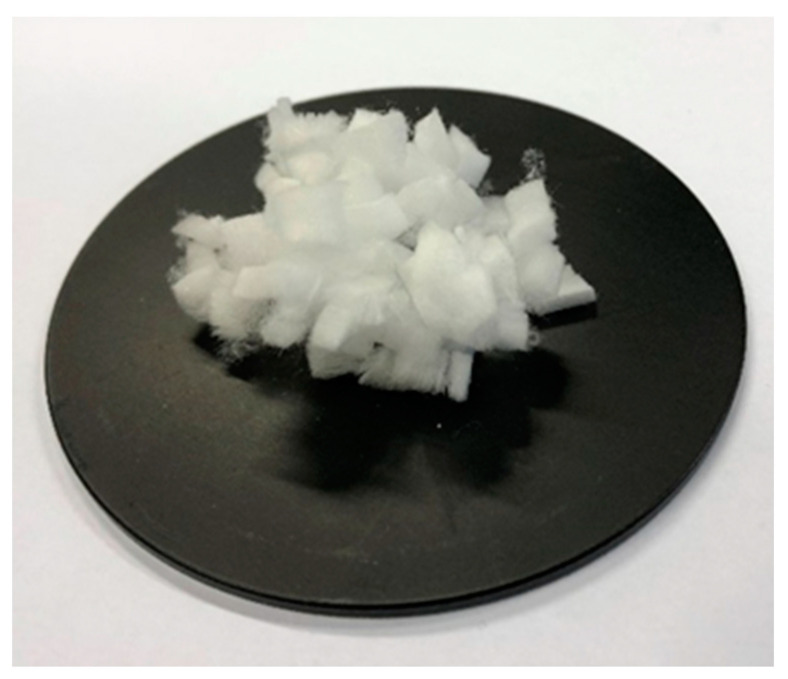
Melamine fibers.

**Figure 2 materials-14-00400-f002:**
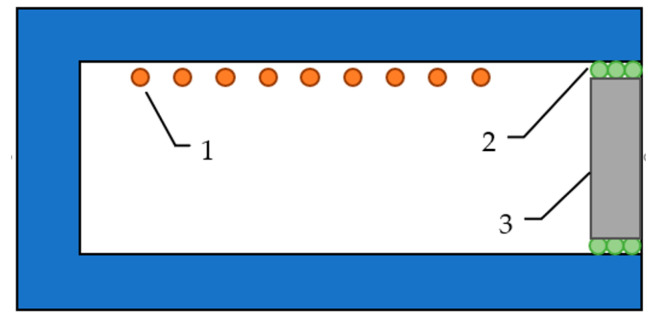
Scheme of the measuring system showing thermal radiation: (1) heating element; (2) high-temperature insulation material; (3) sample.

**Figure 3 materials-14-00400-f003:**
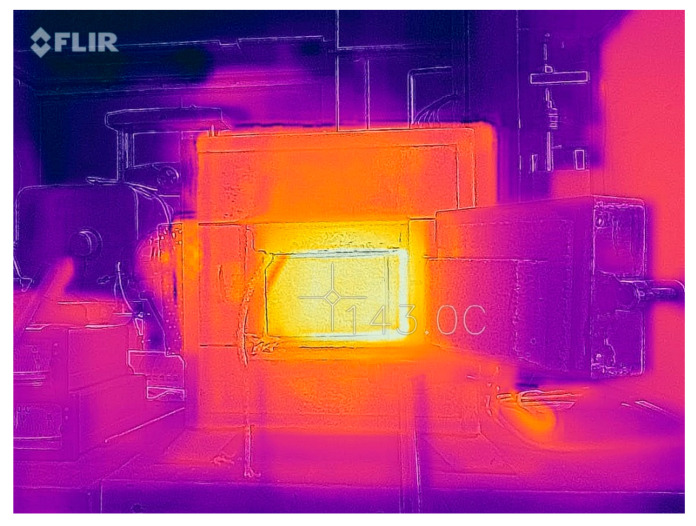
Sample photo of the measurement of thermal radiation.

**Figure 4 materials-14-00400-f004:**
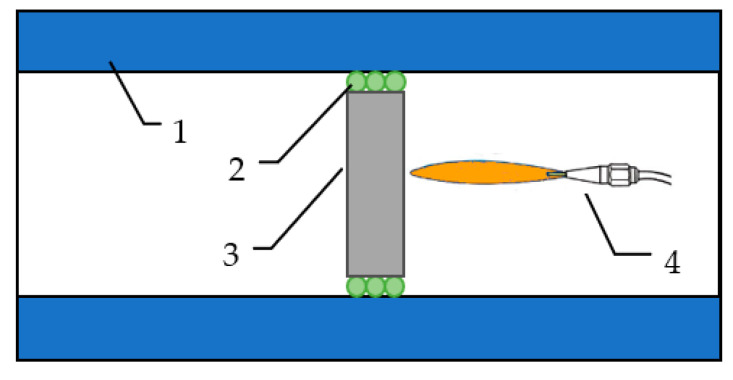
Scheme of the measuring system showing the mini fire-jet test: (1) thermal chamber; (2) high-temperature insulation material; (3) sample; (4) fire-jet system.

**Figure 5 materials-14-00400-f005:**
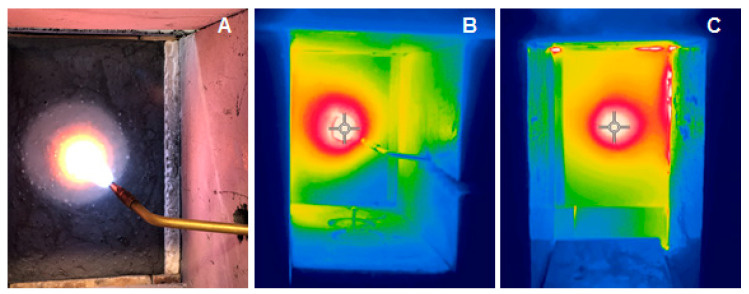
Mini fire-jet test example. (**A**,**B**) View from the side of the combusted gases; (**C**) view of the surface not thermally treated.

**Figure 6 materials-14-00400-f006:**
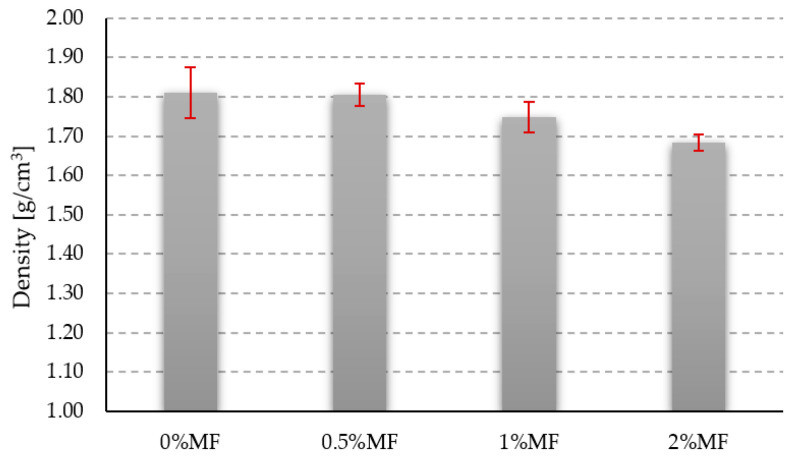
Density results of tested materials.

**Figure 7 materials-14-00400-f007:**
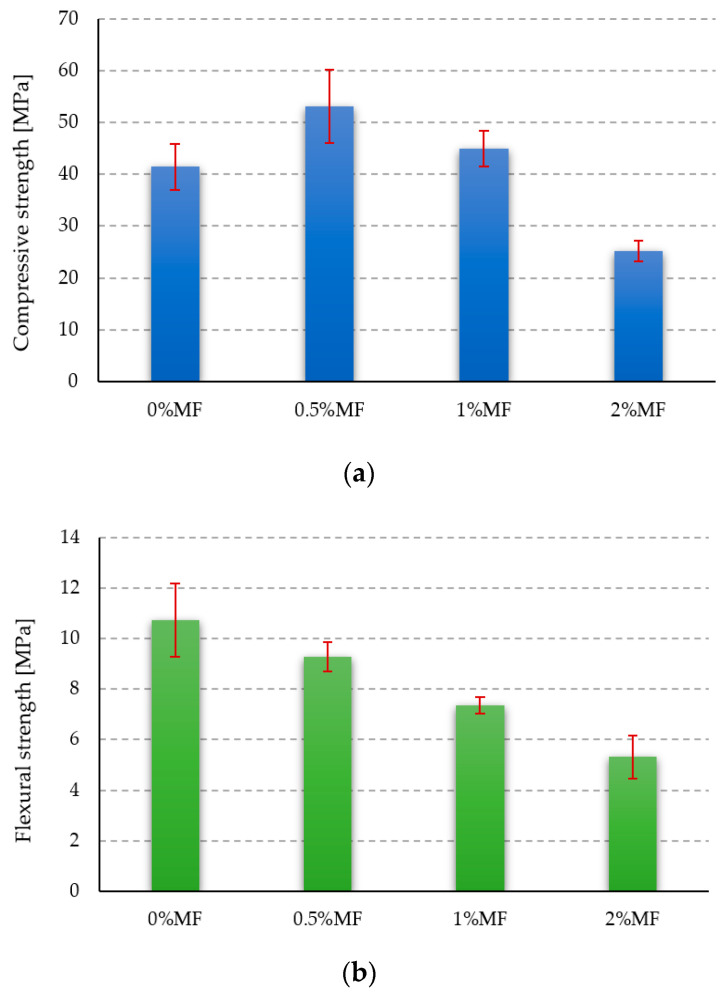
Compressive (**a**) and flexural (**b**) strength of tested materials.

**Figure 8 materials-14-00400-f008:**
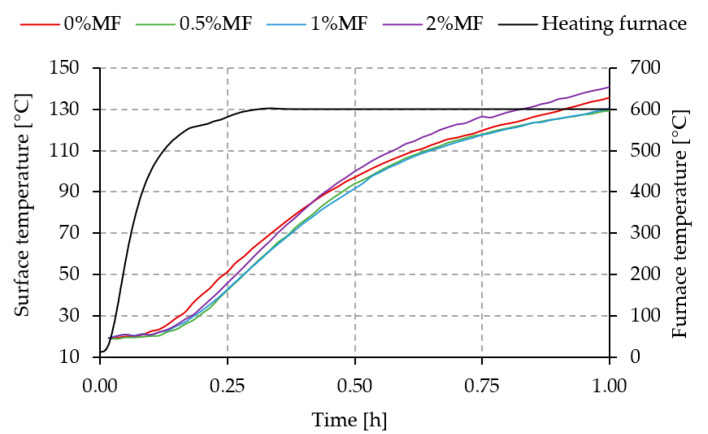
Temperature change depending on the heating time and the melamine content.

**Figure 9 materials-14-00400-f009:**
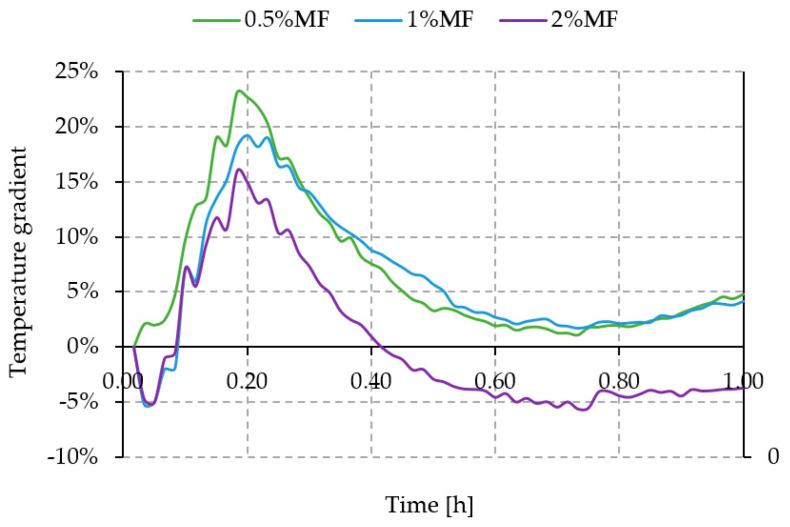
Temperature difference between the pure geopolymer matrix and the geopolymer matrix with variable melamine content.

**Figure 10 materials-14-00400-f010:**
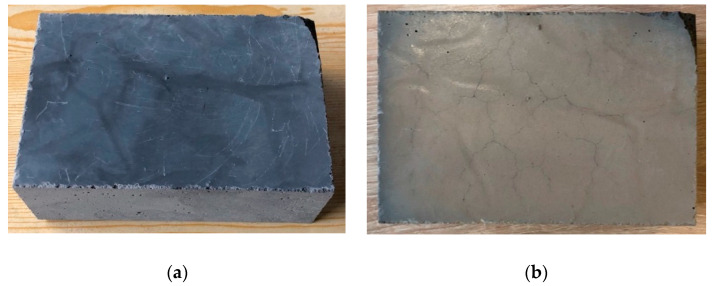
Exemplary photos of the sample surface before (**a**) and after (**b**) exposure to a temperature of 600 °C.

**Figure 11 materials-14-00400-f011:**
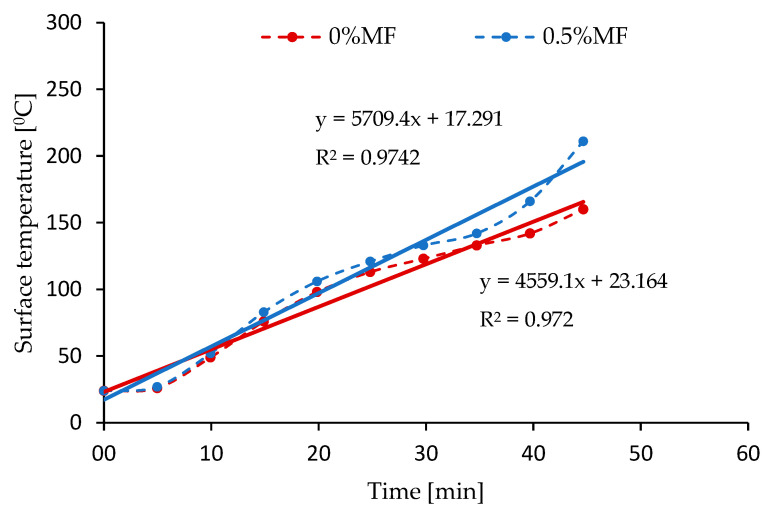
Temperature distribution on the surface of samples during the mini fire-jet test.

**Table 1 materials-14-00400-t001:** Identified phases with their percentage share: XRD analysis results [29].

Precursor	Identified Phase		Percentages Content (%)
	Name	Chemical Formula	
Fly ash	Quartz	SiO_2_	42.3
	Mullite	Al_6_Si_2_O_13_	54.8
	Hematite	Fe_2_O_3_	0.6
	Magnetite	Fe_3_O_4_	0.5
	Anhydrite	CaSO_4_	1.4
	Rutile	TiO_2_	0.4

**Table 2 materials-14-00400-t002:** Designation of the manufactured composites.

Designation	Mixture Proportion (% by Weight)			NaOH Solution
	Fly Ash	Sand	Melamine Fiber	
0%MF	50	50	-	10 M sodium hydroxide solution + water glass (1200 mL in total)
0.5%MF	49.75	49.75	0.5	
1%MF	49.5	49.5	1.0	
2%MF	49.0	49.0	2.0	

## Data Availability

Institute of Materials Engineering, Faculty of Material Engineering and Physics, Cracow University of Technology, Jana Pawła II 37, 31-864 Cracow, Poland.

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
