# Peer review of "Fly-Ash-Based Geopolymers Reinforced by Melamine Fibers"

_materials, 2021, doi:10.3390/ma14020400_

Round 1

Reviewer 1 Report

Materials is a prestigious publication. Therefore it is necessary to improve both the Introduction chapter by indicating some landmarks in the field; state-of-the-art it is necessary to be clear and well defined and argued, as well as in terms of the chapters relating to Materials and methods, Results, Discussions. Generally, a more detailed analysis of the results obtained is necessary: what are the causes, the mechanisms underlying these results, what can be possible solutions etc. by referring to other research in the literature, thus making the connection with the global context of knowledge in the field.

Author Response

Thank you for review and valuable comments. The text have been improved. Thank you for detailed comments in additional file. For the more detailed analysis of the state-of-the-art the new literature have been added, the more detailed information about the material and methods were provided, the more detailed analysis of the results obtained were supplied. The changes and improvements have been marked in red colour.

Reviewer 2 Report

Dear Editor,

The topic of the paper is interesting and suits the Journal of Materials. However, a minor revision is required before this manuscript is qualified to be published in this prestigious journal. The manuscript is needed to be revised grammatically. The authors are required to check the whole manuscript with a grammar specialist as it has several grammatical errors. Only after revising the manuscript based on the comments, the paper is suggested to be published in MDPI. Further information on various issues identified in the manuscript appears below:

  1. The authors have done a great job on the literature review. However, the introduction needs more attention. More information on the new materials:

"Improvement of the early and final compressive strength of fly ash-based geopolymer concrete at ambient conditions." Construction and Building Materials 123 (2016): 806-813.

"Fracture Properties Evaluation of Cellulose Nanocrystals Cement Paste." Materials 13, no. 11 (2020): 2507.

  1. Please provide more detailed reasoning behind the behaviors. The details should include the rigid numbers or percentages. Please add more theoretical discussion.
  2. Add error bars to your figures where applicable.
  3. What is the reason for choosing those W/C ratios and the percentage of SP? Please explain it in the manuscript.
  4. Please indicate how many samples for each experiment have been used.
  5. Please describe the process of each experiment. Also indicate the model of each tool that is used in the experiment. What is the accuracy of each machine? Please explain them accurately.

Author Response

Thank you for review and valuable comments. The English have been improved, including grammar. Below you find summarizing the most important points in manuscript improved according your comments:

  1. The suggested articles are included in revised version of the article.
  2. The additional information have been added to results and discussion parts. The changes and improvements have been marked in red colour.
  3. The error bars are included in figures 6 and 7, the figure 11 includes line of tendency.
  4. The additional information have been added to the section materials and methods.
  5. The informationabout the numer of samples have been included.
  6. The additional information about each test are provided into the text, including accuracy of each machine.The additional figure 4 have been added to article. It clarify the fire-jet test. The changes and improvements have been marked in red colour.
